# A Framework for Analytical Validation of Inertial-Sensor-Based Knee Kinematics Using a Six-Degrees-of-Freedom Joint Simulator

**DOI:** 10.3390/s23010348

**Published:** 2022-12-29

**Authors:** Ariana Ortigas Vásquez, Allan Maas, Renate List, Pascal Schütz, William R. Taylor, Thomas M. Grupp

**Affiliations:** 1Research and Development, Aesculap AG, 78532 Tuttlingen, Germany; 2Department of Orthopaedic and Trauma Surgery, Musculoskeletal University Center Munich (MUM), Campus Grosshadern, Ludwig Maximilians University Munich, 81377 Munich, Germany; 3Human Performance Lab., Schulthess Clinic, 8008 Zurich, Switzerland; 4Institute for Biomechanics, ETH Zurich, Leopold-Ruzicka-Weg 4, 8093 Zurich, Switzerland

**Keywords:** gait analysis, IMU, joint angle, knee kinematics, joint simulator

## Abstract

The success of kinematic analysis that relies on inertial measurement units (IMUs) heavily depends on the performance of the underlying algorithms. Quantifying the level of uncertainty associated with the models and approximations implemented within these algorithms, without the complication of soft-tissue artefact, is therefore critical. To this end, this study aimed to assess the rotational errors associated with controlled movements. Here, data of six total knee arthroplasty patients from a previously published fluoroscopy study were used to simulate realistic kinematics of daily activities using IMUs mounted to a six-degrees-of-freedom joint simulator. A model-based method involving extended Kalman filtering to derive rotational kinematics from inertial measurements was tested and compared against the ground truth simulator values. The algorithm demonstrated excellent accuracy (root-mean-square error ≤0.9°, maximum absolute error ≤3.2°) in estimating three-dimensional rotational knee kinematics during level walking. Although maximum absolute errors linked to stair descent and sit-to-stand-to-sit rose to 5.2° and 10.8°, respectively, root-mean-square errors peaked at 1.9° and 7.5°. This study hereby describes an accurate framework for evaluating the suitability of the underlying kinematic models and assumptions of an IMU-based motion analysis system, facilitating the future validation of analogous tools.

## 1. Introduction

Research suggests that up to a fifth of total knee arthroplasty (TKA) patients are dissatisfied with their joint functionality post-surgery [1,2,3,4,5]. Experts have previously attributed this deficit to the ubiquitous use of standardised treatment protocols across patients, regardless of their specific kinematic characteristics [6]. It has been suggested that the inclusion of gait analysis could improve clinical decision-making and treatment outcomes [7]. Current state-of-the-art technology includes both marker-based and markerless optical motion capture systems, as well as static and moving fluoroscopic systems [8,9]. While fluoroscopy offers the considerable advantage of not being affected by soft-tissue artefact, such systems are associated with substantial processing time requirements, infrastructure, expense and complexity, hence limiting their widespread adoption in clinical settings.

The need for cheaper and more mobile gait analysis solutions has recently been addressed with systems based on wearable sensors, such as inertial measurement units (IMUs). A key hurdle of relying on surface-mounted IMUs to derive joint kinematics is accurately estimating the orientation of the underlying bone segments. This task is made especially challenging by the need to estimate kinematic parameters from linear acceleration and angular velocity data instead of direct three-dimensional positions (as is the case with optical motion capture). Vitali and Perkins [10] recently identified four different categories of methods that tackle this challenge. One of these, the group of so-called “model-based” methods, was highlighted as offering obvious advantages, but requiring more thorough validation before widespread adoption is warranted. Unlike “assumed alignment” methods, model-based methods do not rely on a technician’s ability to visually align the IMU axes with those of limb segments, nor do they expect subjects to properly execute calibration movements, such as “functional alignment” methods do.

An example of a model-based method to estimate the orientation of underlying bone segments has been laid out by Seel et al. [11,12]. Their particular implementation of this method, as later refined by Versteyhe et al. [13], stands out by offering four notable advantages:(1)While numerous IMU-based approaches measuring knee kinematics successfully estimate flexion/extension angles, accurate quantification of ab/adduction and int/external rotation motions are reportedly more challenging. By exploiting a Kalman smoother that leverages two simple mechanical models of the knee joint, this method provides accurate estimates for joint angles in all three anatomical planes.(2)While the user is offered certain guidelines for correct sensor placement, these are minimally restrictive and easy to follow.(3)The approach does not require a separate extensive calibration procedure; there are no additional movements that the patient must perform prior to gait assessment.(4)By avoiding the use of magnetometer data, susceptibility to ferromagnetic disturbances is prevented.

Despite demonstrating promising accuracy in characterising rotational knee kinematics during level walking [13], widespread adoption of the technology in clinical settings remains limited. This is, in part, rooted in the broad spectrum of complex parameter interactions when using IMU-based tools, as well as the associated algorithms that are available. Further variables include sensor placement, pre-processing of raw sensor data, and drift compensation methods [14], to name but a few.

In addition to algorithmic differences, even among systems focused exclusively on capturing rotational knee kinematics, it remains extremely difficult to assess an approach’s accuracy, confounded by contrasting sources of ground truth data. As a result, reaching an objective verdict on how different IMU-based systems perform and which system design is most appropriate can be challenging. Indeed, a systematic review conducted by Pacher et al. [15] reached an analogous conclusion.

### Analytical Validation of Inertial-Based Rotational Knee Kinematics

With the goal of establishing a common foundation for evaluating digital tools that support clinical decision-making, including IMU-based kinematic assessment, the Digital Medicine Society has recently presented the V3 framework [16]. The project lays out standard guidelines divided into three steps: verification, analytical validation and clinical validation. Analytical validation, in particular, focuses on assessing the performance of an algorithm in deriving a physiological metric from sample-level sensor data. The importance of analytically validating IMU-based kinematic assessment tools is therefore clear. Algorithms that derive rotational knee kinematics from inertial data without additional input or extensive calibration generally require the use of models and assumptions. These approximations are needed to compensate for the lack of position data in a global frame, but rarely perfectly reflect reality. Characterising the magnitude of this error can be crucial to guiding product development.

Validation of IMU-based systems directly on human subjects using retroreflective markers and optical motion capture systems [17,18] is difficult due to soft-tissue artefact, whereby the skin on which the sensor is placed moves relative to the underlying bones [19,20]. With known difficulties associated with assessing the magnitude of soft-tissue artefact alone, it is clear that isolating how much uncertainty is due to the algorithm’s models and assumptions, as opposed to soft-tissue artefact itself, is extremely challenging.

Comprehensive analytical validation of an IMU-based system that estimates knee kinematics should therefore occur at two different levels (Figure 1). The first level involves appraising the degree of uncertainty associated with the models, assumptions and approximations that are implemented within the chosen algorithm. In order to isolate this level of error and quantify it experimentally, we require an approach that does not add other considerable sources of error (such as soft-tissue artefact). Current options include the use of manually operated mechanical rigs [21] and industrial robot arms [13,22]. These methods are subject to limitations, such as dissimilar embodiments, susceptibility to easily overlooked errors in kinematic mapping, and difficulties in recreating physiological movements. Thorough testing against references that are able to generate realistic knee motion without soft-tissue artefact are therefore clearly required before system performance can be assessed in real-life scenarios. This investigation consequently aimed to develop and test a protocol for a first-level analytical validation of IMU-based knee rotational kinematics using real joint kinematics reconstructed in a simulator.

## 2. Materials and Methods

To investigate the accuracy of IMU-based systems to estimate flexion/extension, ab/adduction, and int/external tibial rotations, we established a simulator setup that was able to recreate tibiofemoral kinematics throughout in vivo measured functional activities of daily living. To achieve this, we applied the open-source kinematic data collected experimentally by Schütz et al. in 2019 [23]. In their study, a moving fluoroscope was employed to capture the tibiofemoral implant kinematics of six total knee arthroplasty (TKA) patients (average age of 72.8 ± 8.5 years; BMI 24.3 ± 2.2 kg/m^2^), each with a unilateral PFC Sigma cruciate retaining fixed-bearing prosthesis (Depuy Synthes, Warsaw, IN, USA), during level walking, stair descent, and sit-to-stand-to-sit. At the time of assessment, all subjects showed good functional outcome (Knee Injury and Osteoarthritis Outcome Score 91.2 ± 5.7, no/very low pain) and were measured in the gait lab at least 1 year post-operatively (4.2 ± 3.5 years). This subject-specific data was used as input to guide the kinematics of two independent arms of the joint simulator (described below). Output measurements of the physical rotations actually performed by the simulator then served as ground truth data in order to investigate the accuracy of IMU-based angle derivation.

### 2.1. Joint Simulator

A six-degrees-of-freedom hydraulically actuated joint simulator (VIVO, AMTI, Watertown, MA), often used for wear testing applications, was selected for use in this study (manufacturer-reported position resolution: <0.1mm, rotation: <0.1°). Displacement mode and the Grood and Suntay joint coordinate system [24] were used to recreate tibial and femoral kinematics of the robotic segments without the influence of the soft-tissue artefact.

To set the reference pose, the upper segment was positioned to correspond with a flexion/extension gimbal arm parallel to the base plate (Figure 2), and ab/adduction actuators at the centre of travel. Meanwhile, the lower segment was centred along both the anteroposterior (AP; Cartesian Y-axis) and mediolateral (Cartesian X-axis) axes, as well as about the int/external rotation actuator axis. Finally, axial translation was set to correspond with an offset of 0 mm from the default (factory-defined) Z position. These axes corresponded to the original subject data captured in the gait laboratory [23]. This specific configuration was set to correspond to the Grood and Suntay translation (0, 0, 0) and angular coordinates (0, 0, 0).

Iterative learning control was enabled on the AP, flexion/extension and int/external rotation axes (gain fractions of 0.5, 0.2 and 0.5, respectively) to ensure that the simulator kinematic pathway accurately replicated the subject joint kinematics. This iterative learning control feature monitors the tracking error, which refers to the difference between the actuator’s commands and the actual execution. A compensation signal to minimise this error is generated and applied to the following cycle iteration. Here, the goal was for the compensation signal to rapidly reach equilibrium while avoiding dynamic instability, hence allowing accurate kinematic data to be replicated in the simulator. It was found that by the 50th cycle, all input and output simulator signals had converged (tracking error < 1.0°) and become stable. As a result, cycles of each activity were simulated until the kinematic values for the 50th and 51st iterations could be acquired.

Subject-specific kinematic data were made compatible with the simulator coordinate system by expressing rotations of the tibia relative to the femur, as per Grood and Suntay [24]. Standing up and sitting down motions were combined into a single movement by concatenating the respective average signals of each subject in order to ensure a cyclical signal such that the simulator could execute the motion smoothly over multiple iterations. For certain trials in which the subject did not return to their original posture, the kinematic data was smoothed (using De Boor’s approach [25] to fit a cubic smoothing spline with a smooth factor of at least 0.5) to remove any jumps in the signal. Here, smoothing was deemed necessary when the simulator failed to complete haptic mapping (an automated process run as part of the iterative learning control feature [26]). Once the necessary smoothing had been applied, kinematic data corresponding to the subject rotations and anteroposterior translations (mean movement pattern of all activity trials) were input (totalling three activities for each of the six subjects). Imported as text files, data were transferred onto the simulator’s host PC using a USB flash drive. Here, AP translation and int/external rotation were applied to the tibia, while flexion/extension and ab/adduction were applied to the femur. One cycle iteration was performed per second and sampled at 100 Hz.

### 2.2. Inertial Motion Sensors

Two Xsens DOT IMUs (Movella, Enschede, Netherlands) were mounted on the simulator using adhesive strips: one on the upper (representing the femoral segment), and one on the lower (representing the tibial segment) adapter (Figure 2). After time synchronisation of the sensors (Xsens DOT software, v2021.0), linear acceleration and angular velocity data were collected at 60 Hz [27,28] while the simulator performed the kinematic cycles. Data recording was stopped a few seconds after the simulation in order to capture a stationary reference pose.

### 2.3. IMU-Based Knee Kinematics Estimation

A wide spectrum of algorithms to estimate kinematics based on inertial measurements exist [10,14]. Upon review, the algorithm implemented by Versteyhe et al. [13] was selected since this computational method relies exclusively on gyroscope and accelerometer measurements. Correction of drift is achieved using an extended Kalman filter and smoother framework (also referred to as a Rauch Tung Striebel smoother [29]) in which orientation is parametrised as unit quaternions. In general, the proposed set up requires one IMU sensor to be attached to each of the two limb segments (i.e., thigh, shank). The algorithm relies on a rigid body model that approximates the knee as first a hinge and then a spheroidal joint, such that most rotations occur around one main (flexion/extension) axis, while the algorithm still accounts for small rotations in the other planes. Certain kinematic constraints, [11,12,13], are implemented in a first stage to optimise estimates of a joint axis and joint centre of rotation. These approximations are fed into the filter as additional input parameters. Using a pseudo-observation approach [30], a series of output constraints [13] is implemented to update model estimates.

While the original implementation as described by Versteyhe et al. [13] assumes that the IMU sensor frame is aligned with the vertical gravity axis, an additional step was incorporated here to eliminate this assumption by calibrating the gravity vector to a reference static pose [21]. This step allows the user to orientate the sensors arbitrarily on the limb segments. For a given IMU, a rotation matrix describing the relative orientation of the sensor frame with respect to the underlying segment frame is then estimated as follows: The gravity vector during the reference pose defines the proximal axis. Next, the anterior axis is defined as the unit vector perpendicular to this proximal axis and to the main flexion/extension axis of rotation previously estimated as part of the Versteyhe et al. [13] method. Finally, the direction of the mediolateral axis is defined as the unit vector perpendicular to the anterior and proximal axes, hence enabling the calculation of quaternions describing the pose of each IMU with respect to the underlying limb segment.

The original scripts implemented by Versteyhe et al. [13] and Kresie [21] are openly available as MATLAB m-files (The Mathworks Inc., Natick, USA). Custom MATLAB scripts (vR2021b) based on these works were developed, as well as additional scripts for pre- and post-processing of the data. A fourth order Butterworth filter was applied to the raw inertial signals, with a cut-off frequency of 7 Hz. An offset correction script corrected all timepoints to the reference pose orientation. The resulting waveforms were then normalised to 101 points and plotted as a percentage of one gait cycle.

To temporally synchronise the IMU and simulator data, signal alignment parameters (i.e., the time delay between two input signals) based on the flexion/extension waveforms were estimated using the built-in *alignsignals* function and subsequently used to align all three kinematic curves in time.

### 2.4. Data Analysis

Knee joint angles were estimated in the sagittal, frontal and transverse planes, corresponding to flexion/extension, ab/adduction and int/external tibial rotation, respectively. Rotational kinematics were plotted and the root-mean-square error (RMSE) between the sensor-estimated kinematics and the simulator-generated ground truth data was assessed over one representative (51st) activity cycle. In addition to RMSE, the maximum absolute error over the assessed representative activity cycle was also computed to more comprehensively judge the magnitude of errors that could potentially occur at a given time point. Finally, Bland-Altman plots [31] were created to establish the agreement (means and differences) between the IMU- and simulator-based kinematics.

## 3. Results

The collected kinematic data showed the highest levels of accuracy were associated with flexion/extension angles during level walking (Table 1, for subject-specific values see Table A1), with an RMSE below 1.0° for all three rotations (Figure 3). The largest errors in ab/adduction occurred between 60% and 90% of the gait cycle for level walking, and between 40% and 80% for stair descent (Figure 4). In the case of sit-to-stand-to-sit, the highest ab/adduction uncertainties occurred at the beginning and end of the activity cycle, when joint flexion was highest (Figure 5). No clear patterns were recognisable for int/external rotation errors. In general, the flexion/extension rotations were subject to the smallest errors. The largest RMSEs for all three rotations occurred for sit-to-stand-to-sit, with values of 0.9°, 3.1° and 4.8° in the sagittal, frontal and transverse planes, respectively.

Maximum absolute errors for all subjects in the sagittal plane ranged between 1.0° and 2.7° (Table 2, for subject-specific values see Table A2). In the frontal plane, they fell between 0.4° and 7.0°, while errors in the transverse plane ranged from 1.2° to 10.8°. Peak errors for a given plane and activity ranged from 1.6° (sagittal plane in stair descent) to 10.8° (transverse plane in sit-to-stand-to-sit).

Bland-Altman plots revealed that high levels of agreement existed between the IMU and simulator kinematics (Figure A1, Figure A2 and Figure A3). On average, the two approaches differed by 0.5° and presented a mean two-tailed 95% confidence interval of ±2.8°, including results which did not fulfill the assumption of normality (based on a Lilliefors test [32] at 5% significance level). When such cases were excluded, an average difference of 0.2° and a mean confidence interval of ±1.3° was observed.

## 4. Discussion

Validation of IMU algorithms for estimating joint kinematics is critical before sensor systems can be robustly implemented in clinical settings. In this study, we propose and test a framework for validating sensor technology using a joint simulator to assess the accuracy of IMU-derived joint angles. The simulator was driven using subject-specific data to recreate real kinematic profiles, unaffected by soft-tissue artefact. Operation of the robotic simulator using Grood and Suntay, instead of Cartesian, coordinate systems, has specifically allowed simplification of the data harmonisation process and interpretation of the resulting kinematic curves. The simulator approach presented here can therefore act as a reference standard with which to generate realistic kinematics and assess the accuracy of different sensor systems and algorithms to capture real-world movement patterns.

Demonstration of this framework was achieved using a computational approach previously presented by Seel et al. [11,12] and later enhanced by Versteyhe et al. [13]. Our results reveal that this algorithm exhibited high average accuracy (RMSE ≤0.9°, maximum absolute errors ≤3.2°) for all three rotations during level walking. This was in line with previous studies [13] in which IMUs were strapped to two segments of a robotic arm, where the segments represented the femur and tibia. In their study, the femur remained fixed in space, while the tibia was programmed to move. They reported quaternion values of the tibial segment in a global reference frame in a comparison against the robotic arm data. Our framework has been able to build on these preliminary findings by implementing a joint simulator setup in which both femoral and tibial segment analogues rotated (and even translated along one axis). This setup has therefore enabled us to compare the joint angle estimates obtained by calculating the relative pose between limb segments, in a scenario in which both segments rotated relative to a global reference frame. Furthermore, our framework allows the complete three-dimensional reconstruction of measured subject-specific knee joint kinematics. By applying the method of Versteyhe et al. to activities other than level walking, it has been possible to assess the accuracy of IMU analysis during a range of functional activities of daily living.

Based on RMSEs, the IMU-based joint angle estimates were most accurate during level walking. Results during stair descent were slightly inferior, followed by sit-to-stand-to-sit. Similar trends were observed in maximum error values, where level walking was once again subject to the smallest peak errors, while maximum absolute errors during sit-to-stand-to-sit increased to over twice those of other activities. This trend may be a reflection of the algorithm’s underlying assumptions, which are based around applications comprising straight line walking. Here, the joint flexion angles involved in stair descent deviate only slightly from level walking, where both activities exhibit stance and swing phases, with motion advancing in a straight line by alternating movements of the lower limbs. Sit-to-stand-to-sit, on the other hand, lacks a swing phase and does not involve progressive displacement of the lower limbs relative to the ground. This activity is therefore associated with the greatest deviations from the modelling assumptions, hence providing a plausible explanation for the highest measured errors (Figure 3, Figure 4 and Figure 5).

It is clear that out-of-sagittal-plane rotations were subject to the largest errors across all activities. Moreover, visual inspection of the simulator versus IMU kinematics (especially the first two columns of Figure 3, Figure 4 and Figure 5), suggest a correlation between the magnitude of flexion angles and errors in ab/adduction. This association is indicative of cross-talk, which describes how differences in knee mediolateral axis alignment can lead to an artificial increase in the amplitude of ab/adduction angles [33], and additionally propagate to affect int/external rotation angles. Subsequent work should therefore involve a more in-depth assessment of the accuracy of the orientation of the joint (flexion/extension) axis identified by the inertial data, as compared to that defined by the simulator. Furthermore, a more detailed analysis of the nature of the apparent relationship between flexion magnitude and out-of-sagittal-plane rotation errors should be performed. Any such investigation should specifically take into account the problematic lack of consensus surrounding joint segment frame definition.

A review by McGinley et al. [34] concluded that errors in excess of 5° should be avoided in gait analysis to prevent incorrectly guiding clinical decision-making. The advantage of approaching analytical validation of IMU-based gait analysis systems in the framework presented here is that we can conclusively establish whether the underlying technology and computational model for joint angle estimation is able to attain results with mean RMSEs that fall within this 5° window. However, in vivo applications will undoubtedly be additionally affected by soft-tissue artefact, which is likely to compound the model’s uncertainty to exceed this 5° threshold (especially for out-of-sagittal plane rotations in activities such as sit-to-stand-to-sit). Quantification of the errors associated with soft-tissue artefact is beyond the scope of this investigation, but has been addressed by other works [20,35,36]. Minimisation of these uncertainties is known to be challenging and affected by a number of variables, including activity type, movement speed, and subject anatomical characteristics [37]. Nevertheless, the clear advantage offered by the proposed IMU solution is the insensitivity of the approach to sensor placement, which can be leveraged to minimise the effects of soft-tissue artefact [38]. As a result, having established the accuracy of the underlying biomechanical model and its associated assumptions, subsequent work can specifically address soft-tissue artefact and explore promising methods for tackling it.

Although intermittently disrupted by large peak errors (Table 2), the assessed algorithm for IMU-based joint estimation (mainly selected for use in this study due to its flexibility in sensor placement) demonstrated good average accuracy over the assessed activities. Nonetheless, RMSEs of up to 7.5° for individual subjects suggest that the algorithm still requires further enhancement to ameliorate the errors affecting out-of-sagittal plane rotations during activities besides level walking (e.g., by modelling additional constraints imposed by the ankle joint during sit-to-stand-to-sit). Moreover, from a clinical perspective, errors of 7.5° could also critically lead to a different interpretation of joint functionality, particularly if this addresses ab/adduction or int/external rotation. As a result, there is a critical requirement to further develop approaches to mitigate cross-talk in kinematic datasets. Other methods that have attempted to quantify human joint kinematics using IMU technology have utilised different filter implementations [39,40], sensor fusion methods [12,40] and biomechanical models [14] to address drift and attain accurate movement patterns. While the algorithm implemented here is only one of many different approaches to joint angle estimation based on inertial data, the presented framework allows for a standard assessment and comparison of similar solutions.

While the use of a six-degrees-of-freedom joint simulator offered advantages over alternative validation techniques, it was not impervious to limitations. Notably, in a clinical context, both joint segments would be free to move in all degrees of freedom. In the simulator used in our framework, however, all translations, as well as rotations around the longitudinal axis, were performed by the bottom segment, while the remaining two rotations were executed by manipulating the upper femoral segment. These complex kinematic interactions pose a challenge to the algorithm when estimating the motions of the simulator, as only one of the sensors measures active rotational information that is valuable for assessing the model. In a clinical context, the availability of inertial data from both sensors can be fused to produce more accurate joint angle estimates, hence improving system performance rather than hindering it. An additional challenge arose from the boundaries of the simulator’s physical range of motion. For our particular set up and reference pose combination, maximum flexion was capped at 85.0°. During our study, Subject 5 reached 89.4° of flexion during sit-to-stand-to-sit, hence requiring kinematic data to be cropped. However, in general, our framework has demonstrated suitability for recreating real-word kinematic data as a setup for testing IMU algorithm accuracy.

The algorithm for IMU-based joint angle estimation employed in this study relied on Rauch-Tung-Striebel smoothing, an extension of Kalman filtering approaches. Here, filter parameters were defined based on sensor manufacturer specifications and manually tuned by visually inspecting the innovation term for zero mean and no correlation between parameters. In order for the technology to be feasibly scalable, parameter tuning should be performed systematically, repeatably and reliably. As a result, it is crucial that any system that uses a Kalman-type-filter, clearly and transparently reports an associated reproducible method for parameter tuning. This seems by no means to be the norm in the relevant literature published thus far, and is an aspect future investigations are encouraged to address more clearly.

In conclusion, quantifying the inherent error of a given algorithm that estimates joint kinematics from inertial data is a pivotal step in establishing an accurate, yet affordable, and mobile solution to knee kinematic assessment. By quantifying accuracy prior to the introduction of soft-tissue artefact, valuable insights into sources of error can be gained, enabling researchers to better compare the performance of different IMU-based solutions, particularly in light of the challenge posed by cross-talk. The framework presented in this study thus offers a straightforward approach to test inertial-sensor-based systems, and to determine the suitability of the underlying biomechanical models and the limitations of their assumptions. Successful development and adoption of such a system would have widespread clinical implications: By enabling gait analysis to become a regular part of the TKA treatment and rehabilitation workflow, it could help healthcare experts gain insights into the biomechanical mechanisms contributing to patient (dis)satisfaction.

## Figures and Tables

**Figure 1 sensors-23-00348-f001:**
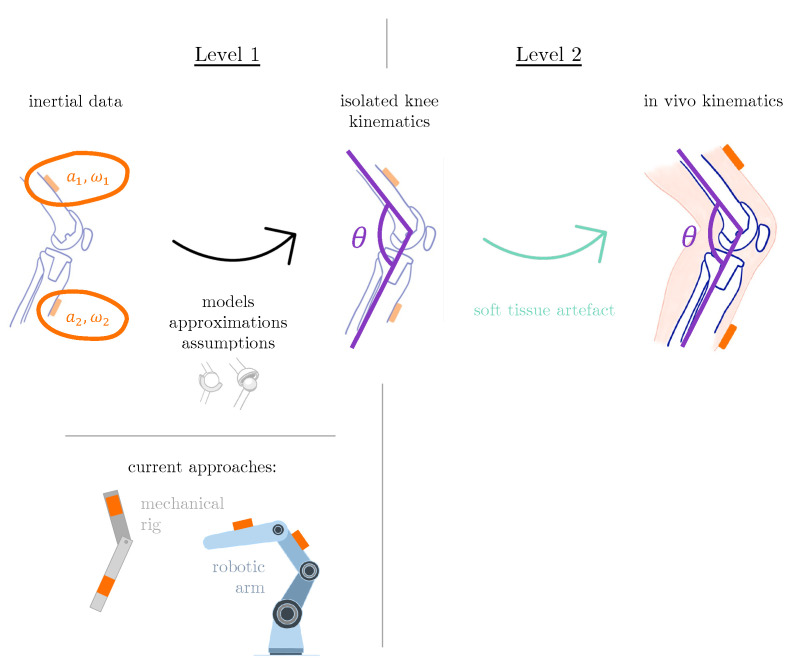
Analytical validation of an inertial-based tool for kinematic assessment should occur at different levels (e.g., with and without the possibility of soft-tissue artefact) to allow characterisation of different sources of error.

**Figure 2 sensors-23-00348-f002:**
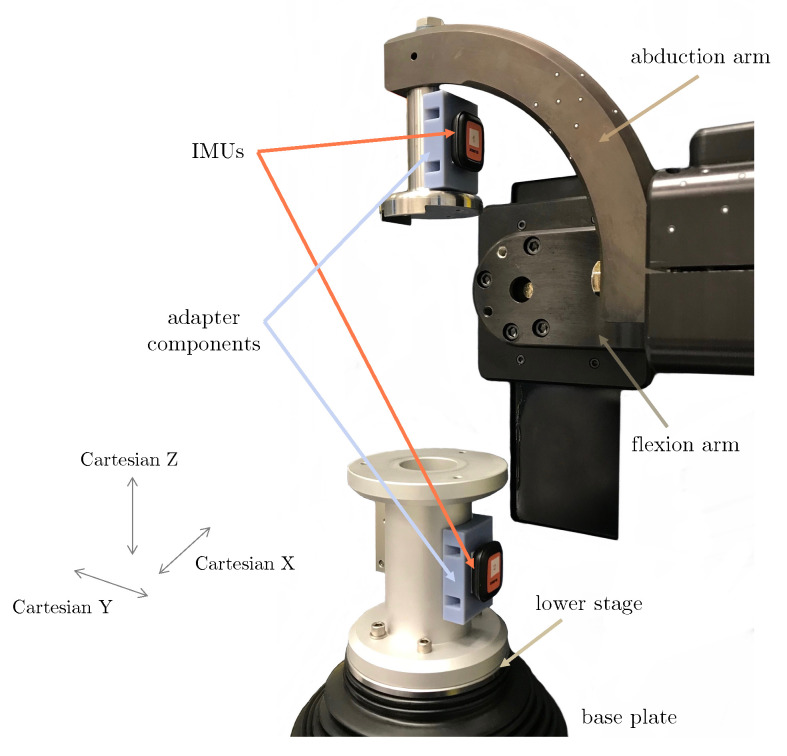
Two inertial sensors were attached to the joint simulator (shown here in reference pose) using mounting adhesive tape: one was attached to the condylar adapter component (top), and the other to the tibial adapter component (bottom). Custom rapid prototype adapters (in blue) were developed to ensure both sensors could be easily adhered onto a flat surface.

**Figure 3 sensors-23-00348-f003:**
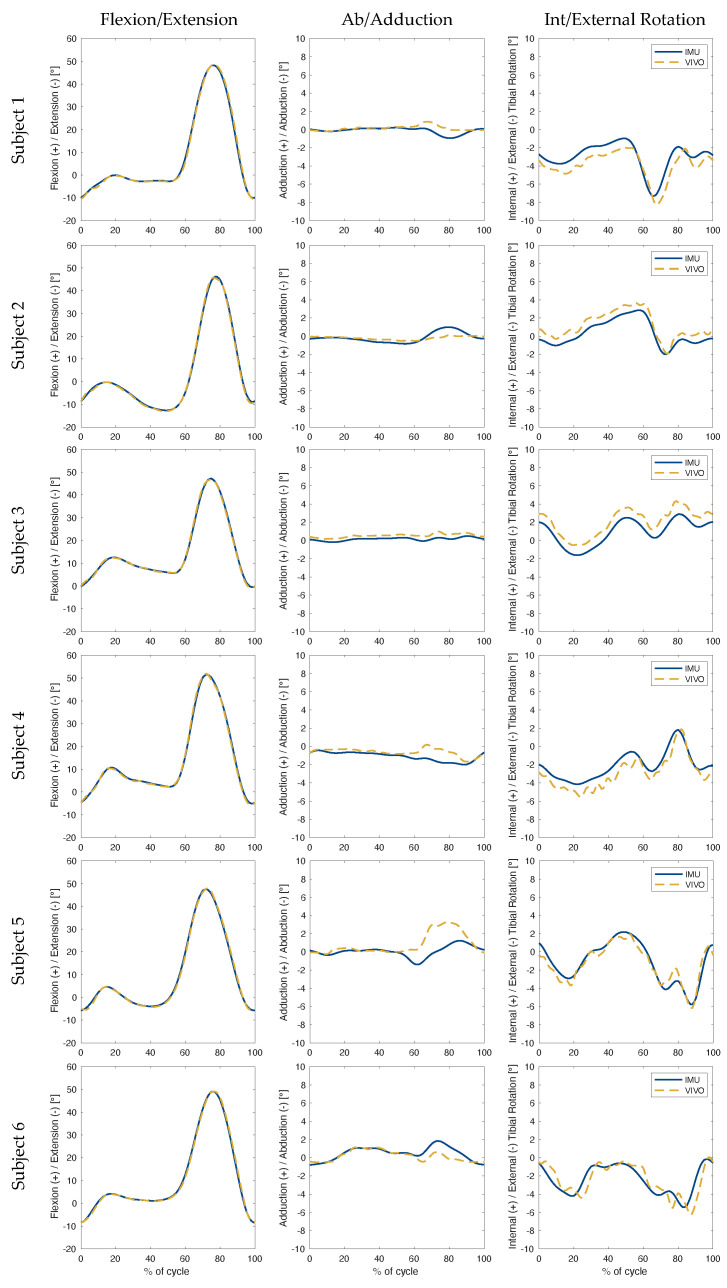
Level Walking: Knee joint angles were plotted over the progression (expressed as a percentage) of one complete gait cycle. The solid blue line illustrates the angles estimated using inertial data. The dashed gold line illustrates the ground truth angles performed by the joint simulator. Each row represents one subject, while each column presents rotations in a different plane relative to the femoral coordinate system.

**Figure 4 sensors-23-00348-f004:**
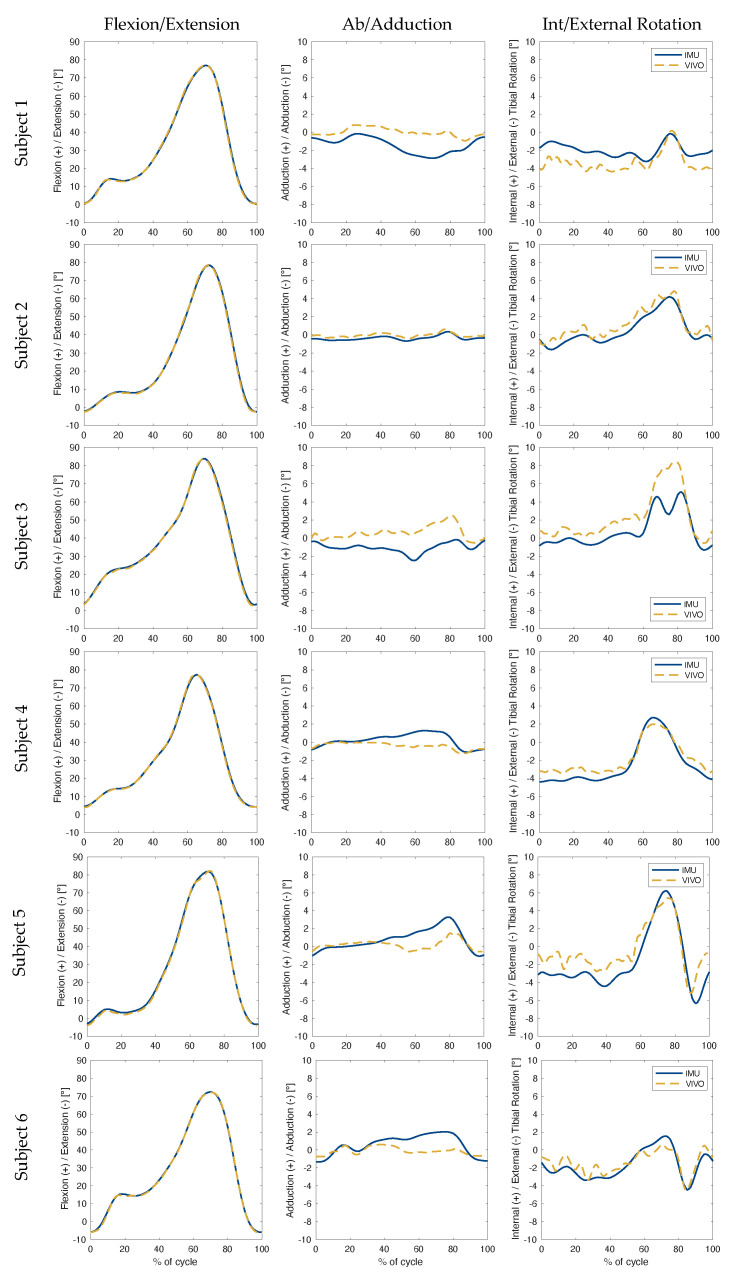
Stair Descent: Knee joint angles were plotted over the progression (expressed as a percentage) of one complete gait cycle. The solid blue line illustrates the angles estimated using inertial data. The dashed gold line illustrates the ground truth angles performed by the joint simulator. Each row represents one subject, while each column presents rotations in a different plane relative to the femoral coordinate system.

**Figure 5 sensors-23-00348-f005:**
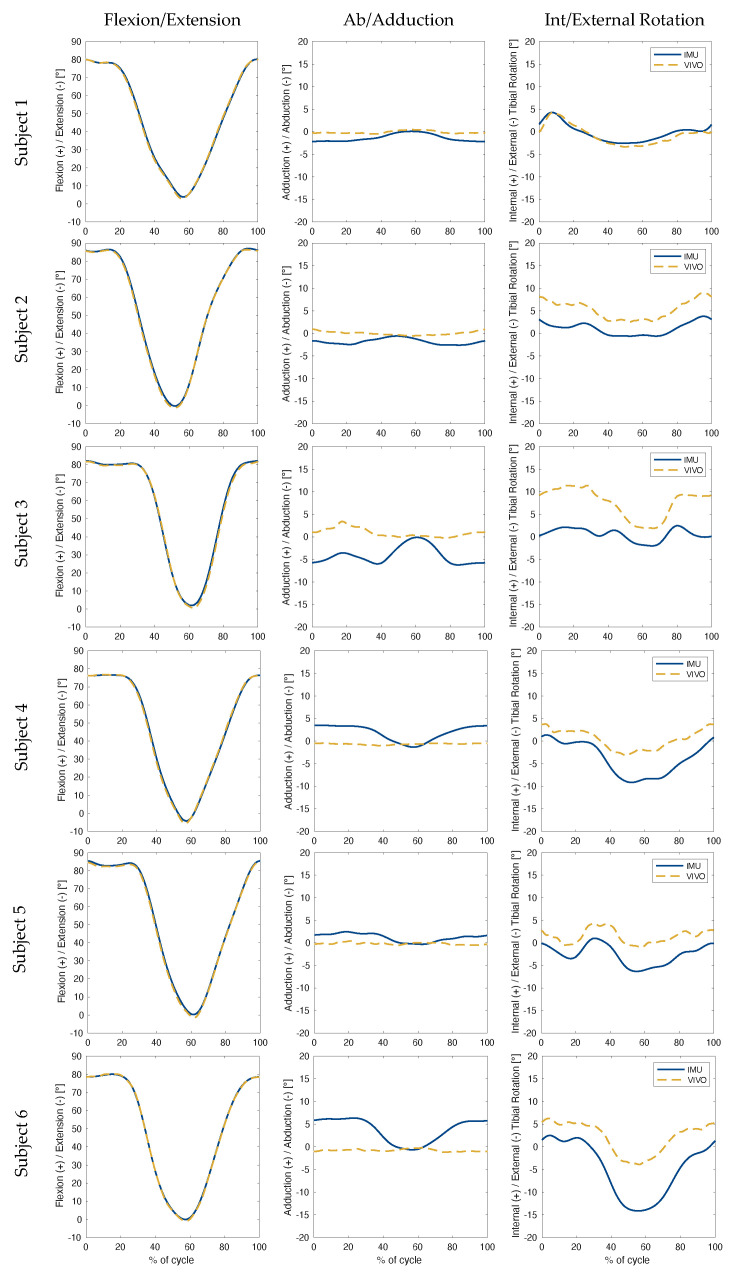
Sit-to-Stand-to-Sit Sequence: Knee joint angles were plotted over the progression (expressed as a percentage) of one complete gait cycle. The solid blue line illustrates the angles estimated using inertial data. The dashed gold line illustrates the ground truth angles performed by the joint simulator. Each row represents one subject, while each column presents rotations in a different plane relative to the femoral coordinate system.

**Table 1 sensors-23-00348-t001:** Root-mean-square error (in degrees) between sensor-estimated kinematics and simulator-generated ground truth data was calculated over one cycle of every activity, for each of the six subjects. Results are summarised by expressing mean RMSE, ±1 standard deviation, over all subjects, for each of the three planes of rotation.

	Flexion/Extension	Ab/Adduction	Int/External Rotation
Level Walking	0.7 ± 0.1	0.6 ± 0.3	0.9 ± 0.2
Stair Descent	0.7 ± 0.2	1.2 ± 0.5	1.3 ± 0.5
Sit-to-Stand-to-Sit	0.9 ± 0.3	3.1 ± 1.9	4.8 ± 2.5

**Table 2 sensors-23-00348-t002:** Maximum absolute errors (in degrees) between sensor-estimated kinematics and simulator-generated ground truth data over all subjects, for every activity type, and each of the three planes of rotation.

	Flexion/Extension	Ab/Adduction	Int/External Rotation
Level Walking	2.3	3.2	2.6
Stair Descent	1.6	3.0	5.2
Sit-to-Stand-to-Sit	2.7	7.0	10.8

## Data Availability

Datasets used for this study are included in Schütz et al. (2019) [23].

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
