# Peer review of "A Framework for Analytical Validation of Inertial-Sensor-Based Knee Kinematics Using a Six-Degrees-of-Freedom Joint Simulator"

_sensors, 2022, doi:10.3390/s23010348_

Round 1

Reviewer 1 Report

In general, the idea and objective of the proposed framework is well described through the paper. The accuracy of a specific IMU Model based on kalman filtering is analyzed using a six-degrees of freedom joint simulator. However, I consider that the two following aspects are not  clear enough: 

1. It is necessary to justify why root mean square errors of 1.9 and 7.5 degrees are not considered as critical.

2. It is necessary more information about the joint simulator, i.e., more technical details about the method used to simulate on it the data acquired from the six people considered in this study. How this info is loaded in this simulator (USB, Wifi, BT)?.  

Reviewer 2 Report

Dear authors,

The manuscript entitled "A Framework for Analytical Validation of Inertial-Sensor-Based Knee Kinematics Using a Six-Degrees-Of-Freedom Joint Simulator" is an intersting work able to be used for reabilitation processes at the knee movement level. The simulation model used was applied to a number of six patiens, the errors obtained being largely acceptable. 

Since the paper is adressed to the Sensors journal and not to a biomecanics journal I propose that some additional figures would be necessary regarding how to use the equipment presented in Fig. 2. Thus, in the Paragraph 2.1 specifications are made regarding the X, Y and Z coordinates as well as the three displacements (0,0,0),and three angles (0,0,0) but these are not indicated in Fig. 2 or in an aditional figures.

Figure 1 is a general figure regarding the posibility to use the thwo inertial sensors on isolated and in vivo knee kinematics.

I hope that some supplementary details focused on the simulator presented in Fig. 2 will improve the paper.
